# Generative Text Steganography with Large Language Model

## ABSTRACT

Recent advances in large language models (LLMs) have blurred the boundary of high-quality text generation between humans and machines, which is favorable for generative text steganography. While, current advanced steganographic mapping is not suitable for LLMs since most users are restricted to accessing only the black-box API or user interface of the LLMs, thereby lacking access to the training vocabulary and its sampling probabilities. In this paper, we explore a black-box generative text steganographic method based on the user interfaces of large language models, which is called LLM-Stega. The main goal of LLM-Stega is that the secure covert communication between Alice (sender) and Bob (receiver) is conducted by using the user interfaces of LLMs. Specifically, We first construct a keyword set and design a new encrypted steganographic mapping to embed secret messages. Furthermore, to guarantee accurate extraction of secret messages and rich semantics of generated stego texts, an optimization mechanism based on reject sampling is proposed. Comprehensive experiments demonstrate that the proposed LLM-Stega outperforms current state-of-the-art methods.

## CCS CONCEPTS

• **Security and privacy → Human and societal aspects of security and privacy**; • **Computing methodologies → Natural language generation**.

## KEYWORDS

Generative Text Steganography; Large Language Models; Black-box; User Interfaces

## 1 INTRODUCTION

Steganography is the science and art of embedding secret information into cover media, aiming to covertly transmit secret information through public channels.[2, 6, 23]. The principle of steganography can be illustrated by Simmons' "prisoner problem"[28]: There are two prisoners Alice (sender) and Bob (receiver) in jail who are trying to hatch an escape plan. The only way they can communicate is carefully censored by warden Eve (steganalyzer). Therefore, they must find some way to embed the secret messages into a normal digital carrier (recalled *cover*) to obtain an "innocent-looking" medium with secret messages (called *stego*). According to the types of digital carrier, steganography can be divided into image steganography [15, 29, 40, 42], video steganography [11, 16, 19, 20], audio steganography [1, 12, 35] and text steganography [7, 8, 21, 38, 44,

*MM'24, October 28 - November 1, 2024, Melbourne, Australia.*
© 2018 Association for Computing Machinery.
ACM ISBN 978-1-4503-XXXX-X/18/06…$
https://doi.org/XXXXXXX.XXXXXXX

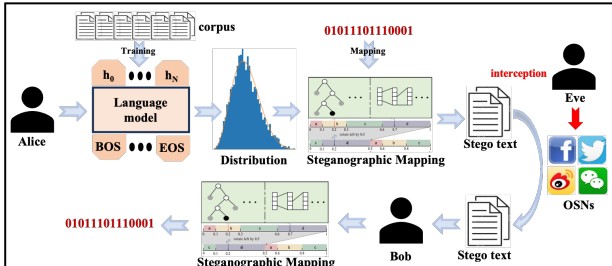

(a) Previous: training language model and designing steganographic Mapping

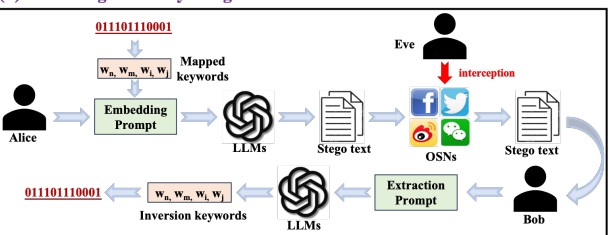

(b) LLM-Stega: directly using the user interfaces of LLMs

**Figure 1: (a) previous methods rely on a language model and steganographic mapping, which is a white-box method; (b) the proposed LLM-Stega generates stego texts by using large language models. The LLM-Stega directly uses the UIs of LLMs to embed and extract secret messages, which is a black-box method.**

46]. Recently, text steganography has become a popular research topic in the information security field since text has high universality and robustness when transmitted through public channels.

Early text steganography methods are mainly achieved by modifying content, such as embedding secret information through synonym replacement [13, 24, 36] or spelling conversion [27]. Although these methods can effectively maintain the semantic imperceptibility of stego text, they have low embedding capacity and significant differences in statistical distribution characteristics [44]. Recent advancements in deep neural networks (DNNs) and natural language processing (NLP) have catalyzed a paradigm shift in text steganography, propelling it towards generative text steganography with high embedding capacity and security. [8, 9, 21, 38, 44–46].

As shown in Figure.1(a), existing generative text steganographic methods typically involve training a language model on a corpus. Subsequently, secret messages are embedded by establishing a steganographic mapping between specific binary bits representing the secret messages and the sampling probability of words within the training vocabulary. These methods have demonstrated commendable performance in terms of security and text quality. However, these methods still have two limitations: (1) These are white-box methods and require that Alice and Bob share the same

language model and training vocabulary. Constrained by the language model and training corpus, the generated texts have a significant gap in fluency, logic, and diversity compared to natural texts. (2) the embedding paradigm, building a steganographic mapping between secret messages and sampling probabilities of the off-the-shelf language model, inevitably changes the sampling probability distribution, resulting in security risk. The first limitation can be resolved by using a superior language model. For example, both Zigegler et al. [46] and Dai et al. [7] leverage GPT-2 as the language model to generate stego text. Recently, Wang et al. [33] use LLaMA [31] to generate stego texts, which is the first attempt for generative text steganography based on LLM. Although these attempts improve the text quality of generated stego text, they cannot achieve satisfied embedding capacity and security since they still use the white-box steganographic mapping to embed secret messages. Some experimental results of these attempts demonstrate the aforementioned second limitation.

In order to address the above questions and fully leverage the generation ability of LLMs, as shown in Figure 1(b), we explore a black-box generative text steganography based on the UIs (user interfaces) of large language models. The contributions of this paper as the following:

(1) It is the first exploration using the UIs of LLMs to implement a black-box generative text steganography, which generates stego texts and extracts secret messages by using some elaborated prompts.

(2) We construct a keyword set and design an encrypted steganographic mapping to embed secret messages. Meanwhile, an optimization mechanism based on reject sampling is proposed to ensure the accurate extraction of secret messages and the rich semantics of generated stego texts.

(3) Comprehensive experiments are conducted to evaluate the superiority of the proposed LLM-stega over the state-of-the-art methods in terms of embedding capacity and security, including Arithmetic codding [46], ADG [44] and Discop [8].

## 2 RELATED WORKS

### 2.1 Generative text Steganography

The main trait of generative text steganography is that an off-the-shelf language model is directly used to generate stego texts under the control of steganographic mapping. Thus, both language model and steganographic mapping play important roles in generative text steganography.

Recently, with the emergence of language models based on DNNs, generative text steganography has continuously made breakthroughs in quality and security. Fang et al. [9] proposed a preliminary idea for generative text steganography. They first randomly divided the vocabulary $V$ into $2^b$ groups $[V_1, V_2, \ldots V_{2^b}]$ to map the $b$-bit secret messages. Then the highest probability token in a group is selected during generation. Yang et al. [38, 39] indicated that using a better language model can generate more fluent and secure stego texts. Both Ziegler et al [46] and Dai et al. [7] use GPT-2 as a language model to generate stego texts. Meanwhile, they designed a different steganographic mapping to embed secret messages. Their findings show that good steganographic mapping

is also beneficial for the security of generative text steganography. From the perspective of provably secure steganography, Zhang et al. [44] proposed an Adaptive Dynamic Grouping (ADG) steganography, which recursively embeds secret information by the adaptive dynamic grouping of the vocabulary tokens. Ding et al. [8] proposed a novel steganographic mapping based on "Distribution Copies" (Discop). Since the steganographic behavior does not destroy the original distribution, this method has state-of-the-art security.

### 2.2 Large Language Models

LLMs have achieved excellent performance on multiple tasks, and researchers have sought ways to utilize LLMs as task-specific data generators. For example, using LLMs to generate tabular data [3], relational triples [5], sentence pairs [26], and instruction data [28]. In these methods, LLMs have a satisfied generation quality for specific subject categories in zero-shot learning. However, existing methods often use simple class condition prompts and some researchers explore ways to improve the quality of the generated data without improving prompt conditions. For example, SuperGen[17] and ZeroGen [41] used LLMs to generate text classification data and used noise robust learning techniques [32] to deal with the quality of the generated data. SunGen [10] used the learned data quality weights to re-weight the data generated in the training process to obtain more excellent data.

Notably, some researchers have begun to explore the use of prompt engineering to tune LLMs and LLMs-API to generate data. Chen et al. [4] explored using soft prompts to tune the data generated by LLMs when white box LLMs and seed samples are used. Yu et al. [43] further proposed a method suitable for black box LLMs and even LLMs-API (for example, ChatGPT) to generate the required data without relying on any labeled samples.

From the above review, it is obvious that existing generative text steganographic methods are the white-box paradigm, i.e., both embedding and extracting secret messages require the off-the-shelf language model and sampling probability distribution of vocabulary on all generation steps. In addition, due to the expensive training and excellent performance, the off-the-shelf large language models have high commercial value. Users hardly access the sampling distribution and rely more on the black-box APIs and UIs to leverage LLMs. Therefore, existing white-box methods are not suitable for LLMs. How to design a black-box generative text steganography with large language models, has become an interesting problem.

## 3 THE PROPOSED METHODOLOGY

As shown in Figure. 2, the proposed LLM-Stega aims to generate stego texts and extracting secret messages by using the UI of LLMs. The LLM-Stega is composed of four parts, including keyword construction, encrypted steganographic mapping, steganographic text generation, and secret message extraction. We will elaborate on each of them in the rest of this section.

### 3.1 Keyword Set Construction

Due to the black-box, we do not have access to the training vocabulary of LLM and its distribution at each step to encode secret messages. Thus, a keyword set is constructed to encode the secret messages, where the keywords are the important components of

**Figure 2: The overall framework of LLM-Stega. Lc-Idx and Re-Idx are the location index and repetition number of the selected keywords in the augmentation keyword set, respectively. Sec-Mes denotes the secret messages. The $w_i$ and $p_i$ denote the i-th word and i-th sampling probability in the keyword set, respectively.**

generated stego texts. In this paper, we select the subject, predicate, object, and emotion as the keywords of each generated sentence. We first design a keyword prompt to induce the LLMs to generate four keyword subsets, containing *subject set*, *predicate set*, *object set*, and *emotion set*. Apart from the emotion set with 3 different words (negative, positive, and neutral) and their sampling probabilities, others contain 16 different high-probability words and corresponding sampling probabilities.

Notably, randomly selecting keywords from the four subsets appear with unclear logic and semantic blur, which brings difficulties to the satisfied stego text generation. In order to mitigate the question, the LLM, induced by an evaluation prompt, is used to evaluate the superiority of the randomly selected keywords and optimize their probabilities. Compared with existing white-box methods, using the constructed keyword set to encode secret messages has three significant advantages: (1) the keyword set is separated from the generation process of LLM, users do not require accessing the sampling distribution of LLMs. Meanwhile, the steganographic behavior does not change the sampling distribution of LLM, which improves the security of the generated stego texts. (2) since the sampling probabilities of the keywords are optimized by using the potential knowledge of LLM, the generated stego text based on these keywords can achieve high text quality. (3) in an ideal situation, the keyword set can encode massive secret messages as long as the set is large enough.

## 3.2 Encrypted Steganographic Mapping

In practice, the number of commonly used keywords is limited and their sampling distribution does not obey uniform distribution. If these subsets are directly used to encode the secret messages, the embedding capacity and security of the generated stego could not be satisfied. To resolve these problems, we first augment the keyword set. Specifically, we perform repeated sampling to expand the keyword set according to the optimized sampling probabilities. Then, the location indices of keywords in the augmentation set are directly used to encode secret messages. In this part, for a good trade-off between time cost and embedding capacity, we expand the capacity of three subsets (subject, predicate and object) from 16 to $2^{18}$ words and that of emotion subset from 3 to $2^{10}$ which can encode $3 * 18 + 10 = 64$ bits secret messages.

In the augmentation keyword set, the more common keyword has a higher sampling probability, encoding more secret messages. The augmentation strategy not only enhances the embedding capacity but also maintains the real sampling probabilities of the keywords encoded secret messages.

Moreover, we notice that the location indices of the keywords are fixed, which leads to the potential exposure risk of the steganographic behavior. To further improve the security of steganographic mapping, an encryption strategy is designed. Concretely, we use a One-Time Password mechanism to implement the encryption. XOR operation is performed using the number of the keyword repetitions and the release time of stego text on online social networks

(OSNs)., which is formulated as follows:

$$S = B_{Re-Idx} \oplus B_{Real-Time} \tag{1}$$

where $B_{Re-Idx}$ and $B_{Real-Time}$ denote the binary of the number of the keyword repetition and that of the release time whose components consist of six numbers, containing date, hour, and minute. $\oplus$ is the XOR operation. The computation result is regarded as a private key shared by Alice and Bob. Although Eve intercepts the augmentation keyword set, he(or she) still cannot correctly decode secret messages.

Compared with existing steganographic mappings, the proposed encrypted steganographic mapping does not destroy the sampling distribution of LLM in the generation process since the steganographic mapping is independent of the generation process of LLM. Because of this trait, the proposed mapping applies to various LLMs with user interfaces. Due to the page limitation, in this paper, we only leverage the UIs of GPT-4 to evaluate the superiority of the proposed encrypted steganographic mapping.

## 3.3 Steganographic Text Generation and Secret Message Extraction

The main goal of the proposed LLM-Stega is that using the user interfaces of LLMs implements steganographic text generation and secret message extraction. After selecting keywords by using the proposed encrypted steganographic mapping, we use an embedding prompt to induce LLMs to generate stego texts. Unlike an explicit extraction algorithm, the extraction based on the LLM cannot ensure a perfectly accurate extraction. Thus, we proposed a feedback optimization mechanism based on reject sampling.

In the generation process, under the guidance of an designed extraction prompt, LLM attempts to output the keywords encoded in the secret messages. If the output keywords have errors, the LLM could return the main reason for these errors and optimize the embedding, generation, and extraction prompts until there are no extraction errors. The feedback optimization mechanism based on reject sampling is shown in Algorithm.1 in detail. In our experiments, making two rejecting sampling ensure the accurate extraction of secret messages embedded into each stego sentence. It is noteworthy that there is a generation prompt behind the reject sampling module shown in Figure. 2. The main reason is that during iterative optimization, the generated stego texts tend to be simple sentences merely containing keywords, so as to ensure accurate extraction. While, the sentence diversity and semantic richness of the generated stego texts are diminished, resulting in secure risk. Therefore, we design and optimize the embedding and generation prompts to ensure rich semantic and accurate extraction, respectively. The optimization mechanism based on reject sampling is shown in Algorithm.1 in detail.

After Bob gains the generated stego texts from OSNs, he (or she) leverages the extraction prompt to extract keywords of the generated stego texts using the UIs of LLM. Finally, according to the shared side information, containing the keyword set, private-key, One-Time-Password mechanism, and the release times of stego texts on OSNs, Bob perfectly decodes the secret messages.

---

**Algorithm 1** Feedback optimization mechanism based on reject sampling.

---

**Require:** Key-words, Embedding prompt, Extraction prompt, Feedback prompt, Generation prompt, LLM
**Ensure:** Stego-text
 1: *accepted* ← False
 2: *Stego-text* ← Generate text carry secret information using LLM by Embedding prompt
 3: **while** not *accepted* **do**
 4:     *Key-words'* ← Extract words from *Stego-text* using LLM by Extraction prompt
 5:     **if** *Key-words'* == *Key-words* **then**
 6:         *accepted* ← True
 7:     **else**
 8:         *Feedback* ← Get feedback for *Stego-text* using LLM by Feedback prompt
 9:         *Generation prompt* ← Optimize Generation prompt using LLM based on Feedback
10:         *Embedding prompt* ← Optimize Embedding prompt using LLM based on Feedback
11:         *Extraction prompt* ← Optimize Extraction prompt using LLM based on Feedback
12:         *Stego-text* ← Generate text carry secret information using LLM by Generation prompt
13:     **end if**
14: **end while**
15: **if** *accepted* **then**
16:     **return** *Stego-text*
17: **end if**

---

## 4 EXPERIMENT

In this section, we evaluate the performance of LLM-Stega in terms of text quality, embedding capacity, anti-steganalysis ability, and human evaluation. The details of the experimental setup and result analysis are described in the following sections.

## 4.1 Experimental Setup

**(1) LLM and Theme information selection.** In the experiments, we select the UI of GPT-4 to implement block-box generative text steganography since the GPT-4 is an advanced and widely used LLM. Due to the vivid topics and popularity, the *Entertainment News* is selected as the theme information to constrain the semantics and context of the generated stego texts.

**(2) Baselines.** Three advanced generative steganographic methods are rebuilt, including Arithmetic [46], ADG [44], and Discop [8]. Due our computation and memory limitation, we use the LSTM model trained on News[30] data as an off-the-shelf language model. For the Arithmetic-based algorithm, we choose the steganography text generated under the different embedding capacities of 1.39 bit per word (bpw) (AC-2) and 3.99 bpw (AC-6) to compare.

## 4.2 Metrics

**(1) Text Quality.** We select the Perplexity (PPL) and semantic similarity (SS) to evaluate the text quality of generated stego texts. For the PPL, it is a general quantitative metric in the other text

**Table 1: Experimental results of PPL and SS**

| Steganography | Perplexity | Semantic similarity |
|---|---|---|
| AC-2 [46] | 199.80 | 0.3421 |
| AC-6 [46] | 287.97 | 0.3723 |
| ADG [44] | 709.84 | 0.4189 |
| Discop [8] | 46.73 | 0.6150 |
| LLM-Stega | 165.76 | 0.5881 |

**Table 2: Experimental results of length and EC**

| Steganography | length | Embedding capacity |
|---|---|---|
| AC-2 [46] | 14.391 | 1.39 bpw |
| AC-6 [46] | 14.944 | 3.99 bpw |
| ADG [44] | 22.411 | 5.63 bpw |
| Discop [8] | 100.000 | 4.76 bpw |
| LLM-Stega | **13.333** | **5.93 bpw** |

generation tasks [18]. The PPL is defined as follows:

$$PPL = \exp\left(-\frac{1}{N}\sum_{i=1}^{N}\log p(w_i|w_1,\ldots,w_{i-1})\right) \quad (2)$$

where $N$ is the length of the text, $w_i$ is the $i$-th token in text, and $p(w_i|w_1,\ldots,w_{i-1})$ is the probability assigned by the language model to the $i$-th word given the preceding words. In this experiment, we choose the GPT-2 model of huggingface to calculate the PPL values of different steganographic texts. For the semantic similarity, We choose the Sentence-bert [25] method and use the *roberta-base-nli-mean-tokens* [14] model to extract sentence vectors, to calculate the cosine similarity between steganographic text and cover text.

**(2) Embedding Capacity (EC).** It is the average number of secret messages embedded into one token, which is represented as bits per word (bpw). It can be calculated as:

$$EC = \frac{N}{W} \quad (bpw) \quad (3)$$

where $N$ is the total number of bits in the embedded secret messages, and $W$ is the total number of words in the generated steganographic text. Since steganography is a key technology for covert communication, embedding capacity is an important metric.

**(3) Anti-steganalysis Ability**. The ability is an important metric for security. In this experiment, we leverage three advanced steganalysis methods, containing LS-CNN (LC) [34], BiLSTM-Dense (BD) [37], and Bert-FT (BF) [22]. The steganalysis accuracy is described below:

$$Acc = \frac{TP + TN}{TP + FN + FP + TN} \quad (4)$$

where $TP$ is true positives, $TN$ is true negatives, $FP$ is false positives, and $FN$ is false negatives, and we assume the stego texts are positive samples.

**(4) Statistical Imperceptibility**. The Kullback-Leibler Divergence (KLD) serves as an evaluation metric to measure the imperceptibility of steganographic algorithms by comparing the distribution of the generated stego-texts against the distribution of the original cover texts. In our experiment, we select the KLD proposed by Zhang et al. [44] to evaluate the statistical imperceptibility of the tested methods. It is formulated as follows:

$$KLD(\mu_x, \sigma_x, \mu_y, \sigma_y) = \sum\left[\log\left(\frac{\sigma_y}{\sigma_x}\right) + \frac{\sigma_x^2 + (\mu_x - \mu_y)^2}{2\sigma_y^2} - \frac{1}{2}\right] \quad (5)$$

where $\mu_x$ and $\sigma_x$ are the mean and standard deviation of cover texts, while $\mu_y$ and $\sigma_y$ represent those of stego-texts.

**(5) Human Evaluation.** For the human evaluation, we focus on evaluating three key aspects of the generated texts: fluency, coherence, and relevance. These criteria are chosen to reflect the intra-sentence linguistic quality, inter-sentence relatedness and causal dependency, and consistency of the generated texts, respectively. We construct a dataset comprising 100 instances of steganographic and non-steganographic texts generated by each algorithm. Three qualified NLP researchers, proficient in English and with a solid understanding of steganography, are employed as annotators. Each annotator independently rates the texts on a five-point scale (ranging from 'very poor' to 'very good') for fluency, coherence, and relevance. The higher evaluation score denotes the better generated stego texts.

## 4.3 Results and Analysis

**(1) Embedding Capacity and Text Quality.** The experimental results of length and EC of generated stego texts are listed in Table 2. To The best of our knowledge, the average length of the News dataset [30] is about 15 words. The Arithmetic and the proposed LLM-Stega have more similar lengths with natural News than ADG and Discop. Meanwhile, the proposed LLM-Stega can achieve the highest embedding capacity. Since the proposed encrypted steganographic mapping is independent of the generation process of LLM, the embedding capacity is not constrained by the distribution entropy of each time step. Table 1 demonstrates the results of text quality of the generated stego texts. In the exploration of literature [39], it is found that texts on public social networks are written by people of different ages and backgrounds in different ways of expression. This leads to the fact that most of the human-written sentences may not obey the optimal language model and form a large variance. From the security, if the generated stego texts have more similar PPL values with normal sentences, their security is higher. In addition, the higher similarity represents the more similarity between cover and stego sentences, namely, higher security. In our experiments, the average PPL of the normal sentence in the News dataset is 185.64. The "AC-2" can achieve the most similar PPL value and the lowest SS. In addition, the "Discop" algorithm gains the best SS and the maximum gap of PPL. While the proposed LLM-Stega can make the best trade-off between PPL and SS.

**(2)Anti-steganalysis Ability**. In the experiment, the training dataset consists of 10,000 cover sentences generated by an off-the-shelf language model without embedding secret messages and

### Table 3: Generated steganographic examples of the proposed LLM-Stega.

| Key-words | Stego-text |
|---|---|
| dancer-announce-song-positive | Acclaimed dancer, Luna Moves, unassumingly announced her new song, which intriguingly entwines rhythmic beats with her signature performative flair. |
| band-praise-song-neutral | The band's praise of their latest song was received with neutrality, a calm before the potential storm of fan reactions. |
| band-released-exhibition-positive | The beloved band joyfully released an immersive exhibition, transporting fans through their rich musical journey, sparking waves of nostalgic happiness across the community. |
| game-released-episode-positive | The innovative game developers ecstatically released a gripping new episode, thrilling gamers and inviting them into captivating new worlds. |
| singer-performed-episode-positive | With a voice that touched souls, the singer performed a memorable episode on the live show, casting a warm, positive light across the enchanted audience. |

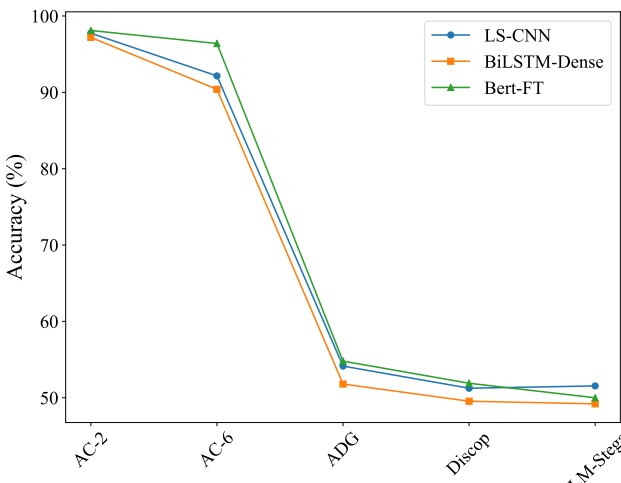
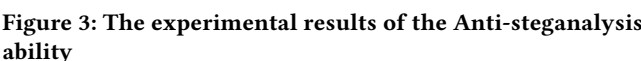
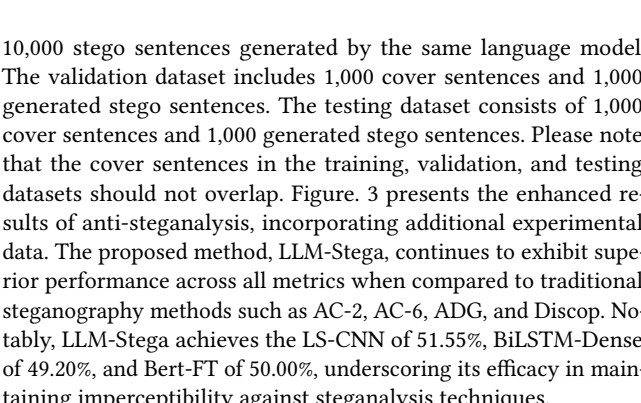

Figure 3: The experimental results of the Anti-steganalysis ability

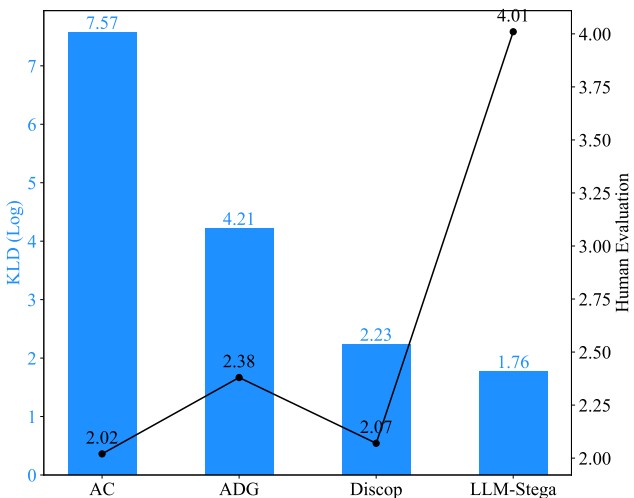

Figure 4: The experimental results of the Statistical Imperceptibility and the Human Evaluation.

10,000 stego sentences generated by the same language model. The validation dataset includes 1,000 cover sentences and 1,000 generated stego sentences. The testing dataset consists of 1,000 cover sentences and 1,000 generated stego sentences. Please note that the cover sentences in the training, validation, and testing datasets should not overlap. Figure. 3 presents the enhanced results of anti-steganalysis, incorporating additional experimental data. The proposed method, LLM-Stega, continues to exhibit superior performance across all metrics when compared to traditional steganography methods such as AC-2, AC-6, ADG, and Discop. Notably, LLM-Stega achieves the LS-CNN of 51.55%, BiLSTM-Dense of 49.20%, and Bert-FT of 50.00%, underscoring its efficacy in maintaining imperceptibility against steganalysis techniques.

**(3)Statistical Imperceptibility**. The KLD thus quantifies the entropy difference between the cover and stego-text distributions, assessing the degree to which the proposed algorithm ensures the stego-texts remain statistically similar to the cover texts.

In this part, we choose the pre-trained BERT to represent the latent feature of cover and stego texts, which is different from the KLD in Zhang et al. [44]. In our experiments, as shown in Figure 4, we find that the probability distributions of the latent features normalized by the softmax activation function obey a normal distribution. We take the logarithm of the experimental results to more clearly compare the differences in statistical imperceptibility between the different algorithms. Among them, the KLD value for the AC algorithm is significantly higher than the other three algorithms, indicating the AC algorithm's poorer statistical

**Table 4: Ablation Study of Prompt Optimization**

| Embedding prompt | Extraction prompt | Text-length | $PPL_1$ | Bert-score | $PPL_2$ | Reject rate |
|:---:|:---:|:---:|:---:|:---:|:---:|:---:|
| ✗ | ✗ | 17254 | 118.80 | 46.23 | 180.55 | 67.80% |
| ✗ | ✓ | 17254 | 118.80 | 46.23 | 178.54 | 42.70% |
| ✓ | ✗ | 12292 | 151.60 | 45.92 | 172.50 | 53.30% |
| ✓ | ✓ | 12292 | 151.60 | 45.92 | 165.82 | 41.40% |
| ✓✓ | ✓✓ | 13094 | 104.67 | 46.33 | 137.26 | 44.10% |
| ✓✓✓ | ✓✓✓ | 20096 | 96.67 | 47.54 | 109.31 | 21.00% |

imperceptibility, which also aligns with its inferior performance in resisting steganalysis compared to the other algorithms. The experimental data confirms that our LLM-Stega method excels in statistical imperceptibility compared to other algorithms, ensuring that the stego-texts are semantically consistent with the cover texts and exhibit similar distributional characteristics.

**(4) Human Evaluation.** In order to further evaluate the effectiveness of the proposed LLM-Stega, we implement the human evaluation experiment. The details of evaluation scores are illustrated in Figure. 4. The experimental results indicate that the LLM-Stega outperformed the other three algorithms. This suggests that LLM-Stega is more effective in generating steganographic texts that are contextually relevant and less detectable. The results demonstrate LLM-Stega's superior performance in generating steganographic texts that are fluent, coherent, and relevant. This finding is significant in the context of text steganography, highlighting the potential of LLM-Stega in applications where the imperceptibility of the embedded message is crucial. Besides, we also illustrate some examples of stego text generated by LLM-Stega in Table 3 for qualitative analysis. We find that the stego text is fluent enough, with correct grammar and coherent semantics.

## 4.4 Ablation Study

As the crux of the proposed LLM-Stega is these elaborated prompts, we carry out the ablation study to verify the effectiveness of prompt optimization. In the experiment, $PPL_1$ denotes the Perplexity score for the initially generated text, serving as a baseline measure of text complexity. **Reject rate** denotes the rate of one rejection sampling, and $PPL_2$ reflects the Perplexity score of the steganographic text following rejection sampling. Furthermore, the **Bert-score** is utilized to quantify the semantic similarity between the steganographic text and the original cover text.

We adopt a structured approach to optimize embedding and extraction prompts, specifically focusing on applications within the steganographic domain. This process is segmented into three phases: an initial optimization phase (denoted as ✓), followed by further optimization phase (denoted as ✓✓), and concluding with a deep optimization phase (denoted as ✓✓✓). "Initial optimization" involved designing the basic structure of text prompts, including theme and a maximum length of generated stego sentences, etc. Based on the initial optimization, "Further optimization" is used to improve text prompts according to the feedback from LLMs. Finally, "deep optimization" consisted of iterative improvements to

the prompt details through continuous feedback, further refining the prompt to achieve the task objectives.

The experimental results are shown in Table ??. We notice that using "Initial optimization" increases the PPL value and decreases BERT scores, namely, the text quality is reduced. This is because the main goal of "Initial optimization" is to ensure accurately extracting secret messages. In subsequent optimization phases, we aim to improve text quality on the condition that the secret information is extracted accurately, achieving high imperceptibility for the stego-text. Thus, both the PPL and the reject rate have decreased. Besides, the single instance of rejection sampling rate is due to the tuned prompt's ability to produce text that allows for 100% extraction of the secret information in just one attempt. The experimental results show that the "further optimization" process meets expectations. After "deep iterations", the probability of rejection sampling drops to its lowest, while the quality of the text is enhanced to its highest. The experimental results demonstrated that the proposed prompt optimization based on reject sampling significantly improved the quality of the generated stego texts.

## 5 CONCLUSION

Previous works of generative linguistic steganography inevitably introduce distortions to the distribution estimated by off-the-shelf language models. In this paper, we attempted to use the user interfaces of large language models for generative-linguistic steganography. Firstly, an encrypted steganographic mapping is proposed to map the secret messages into the words of four keyword sets. These keyword sets are constructed and optimized for the potential knowledge of LLMs. Then, we propose an optimization mechanism based on reject sampling to improve the effectiveness of the prompts. Finally, comprehensive experimental results evaluate the superiority of the proposed LLMs over other tested methods in terms of text quality, embedding capacity, and anti-steganalysis.

The LLM-Stega is the first attempt at generative text steganography based on the UIs of LLM. It is a preliminary method. Meanwhile, steganographic mapping is a simple method to encode secret messages and does not fully leverage the powerful generation ability for text steganography. In future work, we will research a special fine-tuning strategy to further leverage the potential knowledge and generation ability of LMMs for improving the performance of generative text steganography.

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
