# OpenReview forum: "Generative Text Steganography with Large Language Model"
_acmmm.org/ACMMM/2024/Conference — MM2024 Poster_

### Official Review · Reviewer_eZ9a · 2024-05-09

**Rating:** 1
**Confidence:** 4

**Summary:**

This paper proposes a black-box generated text steganography method LLM-Stega based on a large language model user interface. The secret message is embedded by constructing a keyword set and designing a new cryptographic steganographic map. Comprehensive experiments show that the proposed LLM-Stega outperforms the current state-of-the-art methods.

**Strengths:**

LLM-stega claims better than the other methods.

**Limitations:**

Several key weaknesses are brought up in the reviews. This includes the following:
-- It is correct to say in the summary that most users are limited to LLM’s API, but will this make it lower for criminals to pass secret information, making it more difficult to monitor illegal activities and key personnel? In addition, the technical content of keyword mapping and other operations using APIs in the paper is not high, so the practicality of the entire paper is greater than the innovation. Such innovation is difficult to publish in high-level conferences.
-- Although your method has a higher embedding capacity, its perplexity is several times higher than existing methods. Is it reasonable to blindly pursue embedding capacity and give up text quality?
-- The whole work is a bit similar to a codebook, using secret information to find the corresponding keywords, and then letting the API of the large model generate it without going deep into the model. So does this solution itself meet the category of generative? It seems more like a modification.
-- Except for the Discop method, the comparison method is not new enough. Is there no other related work in the past two years? Is the performance of the proposed method brought by LLMs or what? How does the performance compare with work published in the past two years?
-- According to the description in the paper, existing AC and ADG steganography methods can also obtain higher embedding rates by adjusting some parameters. Then the LLM-stega performance claimed by the entire experiment is significantly better than the existing methods, but in fact, there are only 513 lines of “Statistical Imperceptibility” (KLD and human evaluation). This is obviously not enough to support the last sentence in the abstract.
-- In Figure 3, why is the steganographic text obtained by LLM-stega uncharacteristically, and the performance of BERT detection is lower than that of LS-CNN? The reason for this needs to be explained. The detection performance of the LLM-stega method does not seem to be much better than ADG and Discop. So is this result caused by the randomness of deep learning or the superiority of the model structure? This is particularly important.
-- In Figure 4, how are human evaluations performed? Can objectivity be guaranteed?
-- In line 769, appears “Table??”, which seems very unprofessional. In addition, in Table 4, PPL2 is better than PPL1, so rejecting sampling actually reduces the text quality? Can the subsequent operations be completely eliminated and still obtain better results?
-- Is semantic similarity the similarity between texts or the similarity between texts and prompts? The semantic similarity should be very high using LLMs, why is it not shown? Is it because this method of designing keywords is not as good as the methods above in Table 1?
-- The entire architecture seems to make use of LLMs multiple times. Will such a structure lead to extremely high time consumption? In addition, in Figure 2, it seems that LLMs can be used to directly generate Stego Texts, so is the subsequent operation necessary?
If the above questions are adequately answered, I may consider increasing my existing score.

**Suitability:**

2

---

### Official Review · Reviewer_7zXH · 2024-05-13

**Rating:** 6
**Confidence:** 4

**Summary:**

The paper proposes the "LLM-Stega," a novel approach to generative text steganography that leverages the capabilities of Large Language Models (LLMs) in a black-box setting, contrasting with previous methodologies that operated under white-box assumptions. This is noted as the first work to propose a black-box generative text steganography method, effectively utilizing LLMs through their user interface (UI) to generate steganographic text and extract hidden messages. The proposed method first constructs a keyword set and design a new encrypted steganographic mapping to embed secret messages. Furthermore, to guarantee accurate extraction of secret messages and rich semantics of generated stego texts, an optimization mechanism based on reject sampling is proposed. Comprehensive experiments demonstrate that the proposed LLM-Stega outperforms current state-of-the-art methods, including perplexity, similarity, human evaluation, Kullback-Lebler Divergence (KLD) and anti-steganalysis.

**Strengths:**

1.	This paper focuses on the important limitation of existing generative text steganography methods, i.e., existing methods cannot be suitable for the popular LLMs, to design a black-box method. To the best of our knowledge, the proposed “LLM-Stega” is first exploration for black-box generative text steganography, which extends a new research field for the text steganography based on the UI of LLMs.
2.	The experiments are conducted comprehensively and demonstrate the performance of “LLM-Stega” are better than those of current SOTA methods, including ADG and Discop, especially, in terms of the KLD and human evaluation. I believe that these experimental results are convinced. Moreover, this method can implement convert communication with high embedding capacity as long as the keyword sets are large enough, which is suitable for the real-world scenarios.
3.	The English writing of the motivation, challenge, and the method details is clear, and the figures greatly help me to further understand the contributions of this paper.

**Limitations:**

I have a question about the usage of language model in the experiment part. Wang et al., 2024 [1]'s use of LLaMA-2 to generate stego text is mentioned in the text, but this method is not selected as a baseline in this paper. Is there a reason why the authors decide not to include this method in the experiments? In addition, the authors selected a LSTM trained on news as their baseline off-the-shelf language model, which seems to be much weaker than GPT-4. Is there a reason why stronger models are not used as baselines?

[1] Wang, Y., Song, R., Zhang, R., Liu, J., & Li, L. (2024). LLsM: Generative Linguistic Steganography with Large Language Model. arXiv preprint arXiv:2401.15656.

**Suitability:**

2

---

### Official Review · Reviewer_xmg3 · 2024-05-20

**Rating:** 5
**Confidence:** 3

**Summary:**

This work proposes a novel out-of-the-box LLM application for text steganography. The stego-text is first encrypted as sets of keywords. The LLMs are then instructed to generate text with covert messages using the keywords. Also, the authors propose reject sampling so that the stego-text can be correctly extracted from the generated text.

**Strengths:**

* The application of LLMs for linguistic steganography is not well-developed and promising. This work provides a valid method of achieving steganography in the era of LLM.
* The proposed method achieves greater sentence quality and imperceptibility than the previous method.

**Limitations:**

a) The proposed methods are highly limited in their capacity by the keyword sets. The total number of secret bits that could be embedded depends on the size of the keyword set, i.e. $18\times 4 = 72$ bits, as stated in Section 3. As compared with previous works, this is a degradation since they were theoretically capable of attaining an unlimited bit capacity, although in practice, the size may be only a few hundred bits. Also, forcing the keywords to be like *(subject, predicate, object, emotion)* may not be applicable to every scenario.

b) **More experiments on other datasets are required.** Currently, the proposed method has only been tested on one dataset, which raises concerns about its generalizability. It would be better to test the method on more complex datasets, such as IMDB, Twitter and other C4 datasets. Although such dirty datasets may fail the keyword design to some degree, exploring the failure case of the proposed method would also be of value.

c) It is unclear how the KLD is computed in the Supplementary Materials. What is the selected dataset for computing the KLD from the stego-text generated by LLM-stega?

Typos:
* L769, “Table ?? -> Table 4”.

**Suitability:**

3

---

### Official Review · Reviewer_g2MM · 2024-05-22

**Rating:** 4
**Confidence:** 3

**Summary:**

The paper proposed a novel method for generative text steganography, LLM-Stega, that leverages the user interfaces of large language models (LLMs). This is the first study to utilize the user interfaces of LLMs for implementing black-box generative text steganography. The method generates stego texts and extracts secret messages using specially crafted prompts. It introduces an encrypted steganographic mapping and an optimization mechanism to ensure secure and semantically rich covert communication. The method's superiority is validated through extensive experiments, marking a notable improvement over existing techniques.

**Strengths:**

This paper proposed LLM-stega, an interesting steganographic methods via black box large language model. The proposed method is clearly explained. The experimental methodology and results are also well-presented.
This method utilizes words from a LLM-constructed candidate set and employs prompts to leverage LLM for text steganography. This approach simplifies the steganographic process by directly using the user interfaces of existing large models in a black-box manner.
Compared to existing steganographic methods, this approach better ensures the perplexity and fluency of the stego text, making the hidden messages more difficult to detect.
Experimental results show that, compared to the current state-of-the-art steganographic methods, this approach not only ensures the highest level of secrecy but also offers the greatest embedding capacity.

**Limitations:**

1.	From the perspective of motivation, this paper does not provide a detailed analysis of the benefits brought by using black-box LLMs for steganography. The comparison with existing state-of-the-art methods is not entirely fair. For instance, the steganographic process in [8] is provable steganography method, whereas the method proposed in this paper is not provable.
2.	In section 3.3, the reversibility of steganographic information is not detailed. The paper does not explain how the information embedded by embedding prompt can be accurately recovered using the extraction prompt.
3.	The experimental section of this paper primarily focuses on discussing the security of the proposed steganographic method, specifically its ability to avoid detection. It would be beneficial to include some discussion on the time efficiency of this method.
4.	Black-box LLM models, such as GPT-4 used in this paper, introduce randomness when accessed via user interfaces, where the same input might generate different outputs. I am curious whether this phenomenon affects the steganographic embedding and extraction processes. Using an API for steganography might offer more control.
5.	Compared to existing methods, the steganographic approach in this paper leverages more powerful LLM models to enhance the concealment effect of the steganographic text. In comparison, the method in [8], which achieves similar results, only uses GPT-2 as the steganographic tool. Authors may also consider discussing whether employing a more robust white-box model, such as LLAMA 3, in existing steganography methods could potentially yield results more close to those of the black-box GPT-4, while still satisfying provable.

**Suitability:**

2

---

### Meta-Review · Area_Chair_fZet · 2024-06-30

**Recommendation:** Accept (Poster)
**Confidence:** 2

**Metareview:**

This paper received interesting reviews. Three reviewers gave positive recommendations, one very strong too. One reviewer also conveyed strong concerns about the merit of the paper with a strong negative rating. After reading the paper, the rebuttal, and the reviewers comments and discussions, AC thinks that the paper is among many other recent endeavors to use LLM to improve various tasks. However, although AC does not have a problem with the novelty and methodology of the paper, AC is not sure whether the topic of the paper matches with the theme of MM. This is about text steganography, and the approach is purely on the text side, with no other types of input information (e.g., visual, audio, etc.). AC will give a temporary suggestion, and will leave it for senior AC and program chairs to make a final decision.